# (Bio)Ethics in a Pluralistic Society

**Ben Gray** 

Department of Primary Health Care and General Practice, Otago University, Wellington 6242, New Zealand;
ben.gray@otago.ac.nz

**Abstract:** Traditional (bio) ethics relies to a large degree on the analysis of problems to determine the right course of action. In particular, in medicine, a dominant text declares that there is a "Common Morality" that applies to all people. This paper will argue that ethics is culture bound and that, in a pluralistic society, a common morality approach to the resolution of problems has significant limitations. I will argue that more attention needs to be paid to the process of agreeing to a way forward given that there is disagreement. I will illustrate how this applies not only at the clinical level but also at the level of national and international politics. A theoretical understanding of compromise and a look at ways of describing the way people make ethical decisions as opposed to seeking an ideal ethical code is presented as a way in which we can manage problems better in a pluralistic society.

**Keywords:** bioethics; compromise; cultural competence; pluralism; multilateralism

## 1. Introduction

Prescott and Logan [1] argued the need for us to move from the "Anthropocene" to the "Symbiocene" from an era of dominance, exertion of power and competition, to an era of appreciating diversity, multilateralism, and co-operation. This paper will examine some of the tools needed to make this transition. In clinical ethics, the dominant mode has been one of analysis to determine what the right course is, often by those in power. Little attention is paid to how to respond if there is disagreement (other than debating what might be "right"). Such an approach does not work well with an understanding of cultural competence, which acknowledges that different cultures have different values and beliefs. Ethics are culture bound. A similar phenomenon is true in politics at national and international levels where dominant political groups determine what they think is the right course of action and utilize their power to implement it, with scant attention paid to the views of minority cultural groups or nations. This paper will explore the ways in which we might expand both bioethics (as it applies to clinical practice) and ethics/political practice, in recognition of the reality of pluralism at an international level and in many countries at both a national and community level. I will first look at an understanding of culture and cultural competence, and then consider what areas of understanding might be needed to take these factors into account if we want to behave in a collaborative way. I will look at the ethics of compromise, and then consider the distinction between descriptive and normative ethics. I will argue that more emphasis needs to be placed on descriptive ethics while understanding the ethical reasons behind why people disagree, with the hope of making it easier to find a workable compromise. As with Prescott and Logan's paper where they see the microbiome as a metaphor for society at large, I see the resolution of clinical ethical disagreements as an exemplar of the resolution of disagreements at many other levels of society.

## 2. (Bio)Ethics in a Pluralist Society

### 2.1. Clinical Bioethics

Traditional bioethics approaches ethical problems with the general premise that there is a right way (a common morality) to respond to most problems, and that the task of bioethics is to analyse problems to determine what that way is. This is explicitly expressed by Beauchamp and Childress in their text "The Principles of Biomedical Ethics" [2] (p. 3)

*"We will call the set of universal norms shared by all persons committed to morality "the common morality." It is not merely a morality in contrast to other moralities. The common morality is applicable to all persons in all places and we rightly judge all human conduct by its standards.*

A belief in a common morality is congruent with the idea of "doctor centred medicine." This was the traditional approach to medicine where the doctor considered the presentation, analysed the problem, and then decided what the right thing to do was while patients complied.

Where there is disagreement over an ethical topic (for example, euthanasia), bioethics as a discipline tends to provide argument either in favour or against, but provide little other guidance on how to manage the disagreement.

Such an approach could be successful in a culturally homogeneous group of people who have a shared view of what is moral, that aligns with that of the doctor. To consider this, we need an understanding of what we mean by culture, and the related concept of cultural competence.

### 2.2. Culture and Cultural Competence

Ramsden in her work on Cultural Safety defines culture:

*"Being a member of a culture surrounds a person with a set of activities, values, and experiences which are considered to be real and normal. People evaluate and define members of other cultural groups according to their own norms."* [3]

This implies that ethics is culturally bound when mentioning "values that are real and normal." Matsumoto's definition [4] points out that culture is:

*"a dynamic system of rules—explicit and implicit—established by groups to ensure survival, involving attitudes, values, beliefs, norms, behaviours, shared by a group, but harboured differently by each [individual] within the group communicated across generations, relatively stable but with the potential to change across time."*

(p. 24)

The important point here is that cultures are not homogeneous. The values, beliefs, norms, and behaviours are harboured differently by each individual within the group. Generalisations can be made about national groups that have some validity. Hofstede's work of studying cultures has demonstrated that, on six major parameters, you can contrast different countries. For example, Pakistan is significantly more collective compared to the USA, which is more individualistic [5] (p. 95). This does not mean that all people from the USA will be more individualistic than everyone from Pakistan. These values are harboured differently by each individual within the group.

There is a considerable literature that discusses how health professionals should respond to the significant health outcome disparities between cultural groups. Many terms have been coined in addressing this problem: Cultural Competence [6], Cultural Safety [3], Cultural Humility [7], and Cultural Responsiveness [8]. In New Zealand, cultural competence has become the dominant term as a result of the introduction of legislative requirements for all the health professional registration authorities [9].

> *"To set standards of clinical competence, cultural competence, and ethical conduct to be observed by health practitioners of the profession."*

(sec 118(i))

In their policy on cultural competence, the NZ Medical Council defines culture widely [6].

> *"Cultural mores identified by the Council are not restricted to ethnicity, but also include (and are not limited to) those related to gender, spiritual beliefs, sexual orientation, lifestyle, beliefs, age, social status, or perceived economic worth."*

Cultural Humility is defined by Tervalon [7] as:

> *"a process that requires humility as individuals continually engage in self-reflection and self-critique as lifelong learners and reflective practitioners.1-2,7 It is a process that requires humility in how physicians bring into check the power imbalances that exist in the dynamics of physician-patient communication by using patient-focused interviewing and care.8,9. It is a process that requires humility to develop and maintain mutually respectful and dynamic partnerships with communities on behalf of individual patients and communities in the context of community-based clinical and advocacy training models."*

(p. 118)

This provides a good illustration of the ways in which these terms overlap. All of these concepts described under cultural humility are included within the New Zealand Medical Council Policy on Cultural Competence.

My conclusion from this analysis is that, even if the people in a community were in some respects culturally homogeneous (for example, all the same ethnicity, or all adherent to the same religion), they will not all share exactly the same values and beliefs. Every person is a member of several different "cultures." On this understanding of culture, there is no such thing as a homogeneous cultural group.

An understanding of cultural competence was part of the reason for the shift from "doctor centred" medicine to first "patient centred" medicine [10] and more recently "person focused care" [11]. The process of the latter two approaches contrast with that of doctor-centred medicine. They are collaborative negotiated processes dependent on the existence of trust and the relationship.

If, as frequently happens, there is no dispute, the doctor and patient agree on what the problem is and what is the best treatment. It does not really matter which process is used. The outcome will be acceptable. If there is a divergent understanding of the cause of the problem and a difference in what the best treatment is, then a process of working to find a compromise is needed.

An understanding of this has been built into modern medical practice. It is recognised as a central feature of quality practice. Patient Centred Medicine is one of the six aims promoted in the USA to "Cross the Quality Chasm" care that is respectful of and responsive to individual patient preferences, needs, and values, and ensures that patient values guide all clinical decisions' [12] (p. 6). Models of cross cultural care describe how to achieve this [13] and the consultation skills programme used at many medical schools provides detailed guidance on how to achieve this [14]. A traditional approach to bioethics does not sit well with this process.

## 2.3. New Zealand Political Practice

New Zealand has undergone a significant change in the diversity of people in our parliament. This was largely driven by the change from the First Past the Post (FPP) electoral system to a proportional system of electing candidates known as "The Mixed Member Proportional" system. In this system voters vote for a local electorate candidate and for the party they wish to have as government. Each party has a "party list" of candidates (who may also be standing in an electorate) Half of the seats in parliament are filled by electorate candidates. The other half of the seats are drawn

from the party lists such that the final make up of the parliament reflects the portion of the party vote that each party received. Whilst the candidates for the electorates are selected on the basis of who might win a majority in that electorate (and thus favour the dominant ethnicity and historically men), the party list needs to reflect more diversity as every vote counts in determining the final parliament. In the last election, before the electoral change in 1993, only 8% of elected members were from ethnic minority populations (all but one Māori) and 79% were men. In the most recent election in 2017, 34% were from ethnic minorities (Māori Pacific and Asian) and 62% were male [15]. In addition, there are three members identifying as gay and 4 identifying as lesbian (total 5%). Prior to 1993, almost all seats were held by either the Labour (left wing) party or the National (right wing) party. The third largest party in the 1970's Social Credit got 21% of the vote in 1981 but just 2% of seats. Under the MMP system, all parties get the number of seats proportional to the party vote they receive in the election. In the most recent election, the Labour and National parties continued to be the parties with the largest numbers together with a total of 101 seats (84%) and three minor parties were elected. Since 1993, there has always been a coalition government.

Under the previous First Past the Post system, there was always a single party in government and, given that New Zealand is a uni-cameral system, they had the power to implement their programme. The two parties tended to "oppose" each other.

Since the MMP system has been in place, the processes of first forming and then running a coalition government have gradually developed. Our current government is significant in that the Labour party formed the government through a coalition agreement with two of the smaller parties; the Greens and New Zealand First, despite having fewer seats than the National party. Under the previous electoral system, the main focus was on achieving power, which then meant that the policy platform could be implemented with little impediment. Under the new system, there is an incentive for parties to be less oppositional in case they have a need to work as a coalition in the future. A likely reason why New Zealand First chose to go into the coalition with Labour rather than National party could have been the leaking by the National party of adverse private information about the NZ First leader during the election campaign (that he is in the process of suing them about). The policy platform cannot be clearly determined until the detail is sorted between the three government parties. The parties have a high incentive to make the arrangement work and, therefore, the skills of reaching acceptable compromises are to the fore. For the coalition to work, they need to establish functional trusting relationships. Successful leaders under the old system were seen as strong and decisive (and male). Under the new system, leaders need more skills of team building and diplomacy. Since 1993, when MMP was introduced, we have had women prime ministers for 11 of the 24 years.

*2.4. International Politics*

Up until the Second World War, international politics relied very heavily on the exertion of power. The colonial powers dominated large areas of the globe by virtue of their stronger armies, economic controls, and political influence. In his book "Why the West Rules for Now" [16] Morris describes how one response to social pressure at home was to expand, which the British did during their industrial revolution. The British problems of unemployment, overcrowding, and food shortages were able to be addressed with large numbers of people either moving overseas or accessing materials from overseas. This process was dependent upon the use of military power. He goes on to argue that, in our own era, this option for dealing with social pressures is no longer available. There are no more "new frontiers." In addition, he notes that the main problems facing the world are not able to be addressed by the exertion of power but instead require co-operation. He calls them the five horsemen of the apocalypse, which includes global pandemics, famine, refugee movement, failed states, and climate change. These problems can only be addressed by the world as a whole. There is no military solution. We need multilateralism. As our Prime Minister Jacinda Ardern said in her inaugural address to the United Nations.

> *"Any disintegration of multilateralism, any undermining of climate related targets and agreements aren't interesting footnotes in geopolitical history. They are catastrophic . . . together, we must rebuild and recommit to multilateralism. We must redouble our efforts to work as a global community. We must rediscover our shared belief in the value, rather than the harm, of connectedness. We must demonstrate that collective international action not only works, but that it is in all of our best interests. We must show the next generation that we are listening, and that we have heard them."* [17]

The 2017 Nobel Peace Prize recipient Beatrice Fihn addresses the issue of nuclear weapons [18] and pays particular attention to the issue of power and says we need to create a new norm where security does not depend on power. She argues that the whole premise of nuclear weapons is that they act as a "deterrent" by inducing fear of the consequences.

There are some interesting parallels between clinical practice, New Zealand political practice, and an international political practice. The old process relied much more on the main actor (the doctor, the party in power, and the dominant state) having the power to implement what they felt was right. The new processes needed in both a clinical and political practice rely more on developing trusting relationships and being able to find acceptable compromises to move forward.

## 3. An Expansion of (Bio)Ethics

The skills of conventional bioethics/policy development, to analyse and understand problems, to identify what values underpin positions, and to cogently argue in favour of a particular position will always be needed. However, in addition, we need to focus more on how we reach decisions when there is disagreement. Two fruitful areas to look at in achieving this are how to find an acceptable compromise position and ways of understanding why people have different ethical values.

### 3.1. The Ethics of Compromise

What happens next if parties disagree with each other about the right course is usually unaddressed in bioethical discourse. In his paper "On the possibility of principled moral compromise" [19], Weinstock describes three broad ways in which decisions are made as to what to do next.

> *a compromise is a position that, with respect to the issue at hand, is from the point of view of parties locked in debate or negotiation inferior to the positions that both (or all) bring to a decision making process (a negotiation, an election, or more trivially a decision-oriented discussion among friends), but which both have reason to accept instead of the position they favour. They may favour X, when only the issue at hand is in view, but favour Y when all things are duly considered.*

> *The notion of compromise is to be distinguished both from consensus and from what I will here be calling a "settlement".*

> *When a process of deliberation generates consensus, this means that both (or all) parties agree that the position agreed upon is superior to the one they held at the outset, with respect to the issue at hand. (The limit case here is one in which all parties converge upon the position initially held by one party. For that party, the consensus will not be superior, but rather identical, to their initial position.)*

> *A settlement is an outcome that simply reflects the balance of power obtained between the parties. A settlement puts an end to conflict, at least as long as the balance of powers remains unchanged, by allowing the distribution of advantage and disadvantage to determine the outcome. Here, the exchange of reasons that standardly occurs in the attempt at reaching a consensus (or a compromise) does not obtain, or if it does it is not operative in generating the outcome.*

> (p. 539)

Implicit in Beauchamp and Childress's idea of the Common Morality is that the view of a moral person is right and, thus, it is legitimate for them to use the power they have to institute their preferred

outcome by working to achieve a "settlement." In a hospital setting, this might be merely charting the chosen drug without any discussion with the patient (who then acquiesces to it when given). It may be the use of moral authority to coerce patients/parents (in their best interests) [20] (p4). It might be actions such as requiring a patient to sign a "discharge against medical advice" [21]. In paediatrics, the courts can be applied to enforce the clinician preferred treatment against the parents' wishes [22]. If no attempt is made to find out if the patient agrees or disagrees, then there is no possibility of developing a consensus or compromise. The process of informed consent [23] (p. 118) at its best can be a process of reaching an agreed management plan. At its worst, it can be a binary choice with clinician's preferred choice being emphasised. If the clinician is approaching the consultation with a reasonably fixed view of what the best option is, it is hard not to be approaching the consultation more as a "settlement" process and less as a "compromise" process. As Weinstock [24] observes:

> When an agreement reflects a modus vivendi [settlement], it has come about as a result not of the exchange of reasons as to how best to realize some set of values that one feels are at stake, but rather as a result of the balance of forces. A modus vivendi [settlement] therefore, does not embody the values that we have seen inhered in consensuses and in compromises. Parties to a modus vivendi [settlement] are not committed to finding a solution to a problem of collective action that evinces respect or reciprocity.
>
> (p. 639)

This is not to say that the traditional approach to bioethics is never helpful, and that a "settlement" approach cannot lead to a good outcome. On the contrary, in many circumstances, patients will value the opinion of their clinician and be happy to follow their suggestions.

The important point, in this case, is that this model does not provide guidance as to how to address the disagreement. If the clinician proposes a course of action and the patient refuses consent, what should the clinician do? One approach that is the result of using a settlement process is to respond to a patient who against medical advice wishes to go home, by requiring them to sign a document confirming that they were being discharged against medical advice. Such an approach has been advocated to provide "medicolegal protection" in the event that there is an adverse outcome. However, not only does this approach not actually make any difference to legal liability, but the outcome for these patients is significantly worse [21]. If a compromise process were being used then, in the event that the patient and clinician disagree about discharge plans, further discussion is had to determine a discharge plan that both are able to agree to, rather than the physician effectively telling the patient that they are wrong. The development of an agreed management plan in clinical practice, a political compromise at a national level, or a resolution of international problems multilaterally requires focus on the process used. There needs to be a formal deliberation that ensures all values and beliefs are transparently discussed. The consultation framework [25] discussed above that is used to achieve a shared decision in a clinical setting is a good practical example. Weinstock [26] discusses some of the requirements of an effective deliberation aimed at a compromise.

> What determines whether deliberation aiming at compromise adopts a compromise-promoting rather than a compromise-inhibiting direction? Compromise will be made easier where trust exists between the deliberating parties. Trust exists where all parties believe that those with whom they are negotiating are not ill-disposed toward them, or their core interests and values. If I feel as if someone is favourably disposed and respectful toward my most important commitments, I may be disposed to define those core commitments more parsimoniously than if I feel that my deliberation partners are not well-disposed toward me. Now, trust is easier to achieve with respect to a given disagreement where there is already a history of trust between deliberating parties. Conversely, where prior deliberations have given rise to suspicion on the part of the deliberating parties, subsequent deliberations may be more difficult to set on a course favourable to compromise.
>
> (p. 73)

Doctors frequently presume that patients trust them. The preamble to the New Zealand Medical Association code of ethics says "In return for the trust patients and the community place in doctors, ethical codes are produced to guide the profession and protect patients [27]." On a whole community level, this is frequently true. Medical Practitioners were at the top of the group of most trusted professions in a recent survey with a net trust level of 52% [28]. However, there will be many who may not trust doctors, particularly those who have had bad experiences in the past and more likely those from ethnic minorities. We need to pay more attention to levels of trust.

In brief, compromise requires trust, respect, reciprocity, and the absence of coercion. This is not to say that deliberating to find a compromise is easy but, for all the areas we are discussing, it is the most promising way of finding lasting approaches. As Weinstock notes above, an approach based on "settlement" will only last as long as the balance of power remains unchanged. If the patient leaves the hospital, they may stop taking medication started in the hospital. A coalition government is vulnerable to changes in the popularity polls of its constituent members. Nation states will determine their trade relations not on an agreed upon rules-based system like the World Trade Organisation, but on the basis of the extent of their political power, which is currently playing out in the trade war between the USA and China.

### 3.2. Descriptive Ethics versus Normative Ethics

Beauchamp and Childress distinguish between normative ethics and descriptive ethics below [2].

*General normative ethics addresses the question "which general moral norms for the guidance and evaluation of conduct should we accept and why?" . . . descriptive ethics . . . is the factual investigation of moral beliefs and conduct. It uses scientific techniques to study how people reason and act.*

(p. 1–2)

The focus of their text is to determine normative ethics.

*"Descriptive ethics . . . [are] nonnormative because their objective is to establish what factually or conceptually is the case, not what ethically ought to be the case."* [2] (p. 2)

In espousing the idea of a common morality by corollary, they are arguing against "Cultural Relativity" [29]. They argue that saying that a particular practice is acceptable because it is a "cultural practice" is untenable and that we should be able to determine practices that are unethical in all settings. The difficulty with this argument is that any practice happens within a culture. I do not discount the possibility that there might be universal ethical principles such as a normative ethic. However, on a practical level, it is most likely that neither party to a problem are an exemplar of the ideal. The task of trying to determine what a Common Morality or Universal Human Right might be is vital. We need to debate what our aspirational ideal is. However, in resolving disagreements, there is insufficient agreement on such universal codes for them to always be helpful in achieving resolution.

On a national level, laws are passed that determine behaviour on ethically contentious issues such as abortion, euthanasia, and blood transfusion of Jehovah's Witness children against their parents' wishes. While this provides some certainty, the problem of this approach is that the law reflects the values of those who pass the law. Thus, while governments were comprised predominantly of men, abortion was illegal. The frequency of illegal abortions underlined the lack of community consensus on this issue and, around the world, it has been found that making abortion illegal is ineffective in stopping abortions.

Insufficient attention has been paid to descriptive ethics. In resolving real world disagreement, it is often not helpful to know what people think they ought to do and much more useful to know what they will do and why.

A clinical example illustrating these ideas is the case of a 38-year-old Māori woman admitted with vaginal bleeding. Investigations were done and the registrar met with her to discuss the findings

at 11:00 p.m. He explained that she was pregnant but that the pregnancy was in the fallopian tube, that the pregnancy would not continue to term. He had booked theatre for the next morning to remove the tube and pregnancy. Otherwise, it would rupture. The woman responded that she was trying to get pregnant and was pleased to hear she was pregnant and not sure she was ready to make the decision to have it terminated. In addition, she had to be in court the next morning to support her nephew who was charged with assault and she was the only family member who was going to be there to support him. The registrar was under significant time pressure and was frustrated that she would not take his advice. He got her to sign a "discharge against medical advice" form and she went home. She was admitted three days later to the Intensive Care Unit in shock from a ruptured ectopic pregnancy. Instead of continuing to work to achieve an agreed management plan, the registrar tried to coerce her into following his plan (by getting her to sign) and was not prepared to consider any other management that would take into account her wishes. She had the power to leave and he had no power to detain her. If, instead of approaching this using a "settlement" process, he had been approaching it looking for an agreed upon compromise, the registrar would have attempted to respect and acknowledge the importance of her going to court. A clinically appropriate compromise would have been to say to her that he was really worried about the possibility that, if the pregnancy ruptured, she could become very ill, and to arrange a time to talk after the court case including giving her a phone number to ring in the event that she had increasing pain, any dizziness, or palpitations.

The fact that the registrar knew what should happen did not help achieve the best outcome. If, instead, he had been interested in why the patient did not agree with him, and had been able to acknowledge that the proposed course of action was negotiable, it could have led to a better outcome. In retrospect, she clearly needed the operation, but it could have been done the next day after the court case had been completed. There may well have been existing low levels of trust by the Māori woman in the health system that would have been made worse by a feeling of not being listened to (and acknowledging the importance to her going to court) and significantly disrupted by the requirement to sign the form. The loss of trust may well have delayed her re-presentation in the hospital when her condition deteriorated. While it could be said that the outcome was her responsibility alone, it is very likely that issues of implicit bias [30] if not overt racism contributed to this outcome.

In trying to find a way through a disagreement, an understanding of the context of disputed values may enable us to find a way forward. I will discuss two approaches to this, first, by understanding different cultural approaches to ethics, and, second, by considering ethics as it might apply to people at different levels of Maslow's Hierarchy of Needs [31].

### 3.3. Respecting People Who are Different from Me

An important prerequisite of a deliberative process to find a compromise is for all the parties to respect each other. This sounds straight forward, and is frequently not a problem for me as a doctor when I am caring for people "like me." However, the reason for the development of the concept of Cultural Competence was that people from non-dominant cultures have worse health outcomes [32], and one element of why that happens is that they are not respected as equals. Addressing this requires considerable insight by the clinician into their own culture. Part of being a member of a culture is that those you share the culture with are "in your group" and others are "out." Your beliefs and values are normal and right and those who do not share them are, at best, different but, more problematically, wrong. Many of our beliefs and values are held implicitly, and may, in fact, clash with our espoused views. The Harvard Implicit Association Test [33] has convincingly demonstrated that, as a result of years of acculturation, the large majority of people worldwide associate black people with unpleasant associations more than white people. The negative effects of this can be mitigated by having insight into one's own culture and the way that it shapes views on the world. On a clinical level, it requires consciously suspending judgement while the issue in question is explored, rather than allowing an unconscious bias to determine the outcome.

A good example of this would be the manner in which allopathic doctors react to homeopathy. The "Culture of Medicine" [34] is based strongly on the idea of evidence-based medicine and espouses the view that practices that are not evidence-based are unethical [35]. However, if the issue is examined in more detail [36], it is clear that most of the conventional medicine is far from evidence-based. Surgery, in particular, rarely has a high level of evidence to justify its practices [37] and, unlike homeopathy, always causes harm. We have a limited understanding of the placebo effect, which means that many people get better with treatments for which we have no evidence because of their belief that they will help. If the doctor were to be disparaging (disrespectful) of a patient who uses homeopathy, then it is less likely that an effective trusting relationship will be able to be developed. This does not mean that the doctor has to agree with the use of homeopathy. There are certainly clinical circumstances (where conventional treatment has a high chance of success for a serious condition) when it would be a doctor's duty to disagree with that approach. This can be discussed without disrespecting the person and, in fact, a better outcome is more likely if a respectful dialogue can be held discussing various options.

### 3.4. Spiral Dynamics and the Hierarchy of Needs

Any judgement on the morality of behavior must take into account where, on the hierarchy of needs, the person may sit. I have met a Somali refugee who told lies about his identity while in the refugee camp in Kenya, and, as a result, was accepted as a refugee into New Zealand. Now in New Zealand, where he is safe and well fed, lying cannot be condoned, and he faces a difficult problem of whether to correct the lie at the risk of being deported. It is hard to argue that lying is always bad if there is a choice between staying in a refugee camp with all the risks of starvation, disease and physical as well as mental trauma, and lying to get out. However, the fact that this lie continues to have consequences illustrates the dynamic nature of some ethical choices. The other party to this ethical decision is the New Zealand immigration service. They wish to run a fair system of selection of refugees and, if all refugees lied, then this aim would be subverted. This illustrates the relevance of understanding the reasons behind a particular act. There remains a judgement to be made, by the man as to whether to now tell the authorities the truth, and, if he does tell the truth, by the authorities as to how to respond to his admission. He is faced with a problem of how to maintain his own integrity and coherence in the new setting. Ethical behaviour differs depending on where a person is on Maslow's hierarchy of needs [31].

In the 1970s, Clare Graves developed the idea of Spiral Dynamics [38], which is an extension of Maslow's work. It is a psychological theory of values where he argues that people both individually and collectively develop up the spiral. Table 1 describes five of his levels [39].

**Table 1.** Conditions and Coping Mechanisms utilised at each level of Psychological Existence.

| Graves Level of Psychological Existence | Life Conditions | Coping Mechanisms |
| --- | --- | --- |
| Level 3 Red (c-p) | egocentric. power/action, asserting self to dominate others, control, sensory pleasure | Breaking away from the tribe, impulsive, seeking respect, honour, and avoiding shame and establishing the self, might is right. The world is adversarial, uncaring, only raw power will let me prevail. |
| Level 4 blue (D-Q) | absolutistic: stability/order, obedience to earn reward later, meaning, purpose, certainty | Emerges from the chaos of C-P—obedience to rightful authority, binary thinking, categorising, deny self for 'the one right way,' stability and security is achieved through sacrifice and submission, doing things by the book/manual, bringing in new norms undermines control/authority. |

**Table 1.** *Cont.*

| Graves Level of Psychological Existence | Life Conditions | Coping Mechanisms |
| --- | --- | --- |
| Level 5 ORANGE (E-R) | multiplistic: opportunity/success, competing to achieve results, influence, autonomy | Emerges from the rigidity of D-Q, how to manoeuvre rather than comply, many ways and criteria rather than one right way or set of standards, goal directed, independent, self-sufficient, confident, experiment to find the best among many possible choices, future oriented and competitive. Working for a good life and abundance, the winners deserve their rewards. |
| Level 6 GREEN (F-S) | relativistic: harmony/love, joining together for mutual growth, awareness, belonging | Emerges in response to the excesses of E-R, cannot do it on my own and need to collaborate with others, group membership highly valued, tolerates ambiguity by encountering diverse perspectives, requires trust, does not want to hurt others, high empathy and sensitivity to others – everybody counts. |

These levels are particularly useful when looking at international relations. Donald Trump clearly operates from around level 3.

*Impulsive, seeking respect, honour, and avoiding shame and establishing the self, might is right. The world is adversarial, uncaring, only raw power will let me prevail.*

Jacinda Ardern is more at level 6:

*cannot do it on my own and need to collaborate with others, group membership highly valued, tolerates ambiguity by encountering diverse perspectives, requires trust, does not want to hurt others, high empathy and sensitivity to others – everybody counts.*

For those who disagree with President Trump, the strategy of arguing the merits of his policies has proved a particularly ineffective one. He believes that what he is proposing will make America great, but it is coming from a psychological position that does not easily collaborate with others and believes that the exertion of power will solve all problems.

Applied to an individual clinical setting is the example of Oshin Kizsco [40], which illustrates some similar points. Oshin had a brain tumour and had suffered significant side effects from the initial surgery. The doctors recommended that he have chemotherapy, but, despite knowing that without this, he was likely to die within three months, and with it the five-year survival was 50% to 60%, his parents declined chemotherapy.

The parents raised concerns about the impact of Oshin's future care on them as his mother suffered chronic back pain and was limited in her ability to care for Oshin and his father was concerned about potential financial implications of being required to take unpaid leave to care for his son if required [40] (p. 141).

In discounting the relevance of these concerns, the doctors were presuming that "survival" was assured and higher level considerations of extending life clearly outweighed these considerations. The parents were (in part, there were other arguments) putting the case that they were not sufficiently financially or socially secure to be able to sustain the care that Oshin might require.

This analysis is also useful in looking at responses to youth offending in New Zealand. By definition, children are going to operate from a lower level. In a recent documentary discussing secure residencies for youth offenders, the case leader at Korowai Manaaki, Luke Wilson says:

*most of the young men who come into this unit arrive with a host of problems: trauma, gang connections, developmental delay, lack of education, poor communication skills, and drug and alcohol addictions.* [41]

Expecting these young men to make "good choices" is unrealistic and understanding them as people who are still learning and responding to survival needs rather than being bad is more likely to lead to better outcomes.

An understanding of this culture, is about starting from a position of respect. If there is a disagreement, then the other party has good reasons for disagreeing with you, and a better understanding of those reasons will increase the likelihood of finding a compromise.

### 3.5. Values and Traditions of Minority Groups

A common response to the approach described above is to raise the issue of minority practices that "we" find abhorrent. Parekh's analysis of how we, as a society, should address these issues provides a philosophical framework to support the approach I am advocating [42]. A commonly cited clinical example would be the practice of female genital mutilation. Macklin [43] discusses this example and others in some detail. As part of the discussion, she describes an episode at Harbourview Medical Centre in Seattle where a family requested circumcision of their daughter. I understand that this came about because the child was admitted for circumcision and it was only when seen in the clinical practice that the gender of the child was confirmed to be female (the doctors were expecting a boy). The doctors clearly had a respectful negotiation and came up with a compromise that they and the parents were happy to accept, of performing a ritual nick without removing any tissue. This led to a furore with much public comment and the approach was abandoned by the doctors. In my view, this is a shame and hard to justify ethically. The surgeons were, after all, planning to perform a much more mutilating procedure if the child had been a boy. There has been a reaction of moral panic in western countries to the practice of female circumcision. Calling it genital mutilation will mean that any discussion with a family wishing this for their daughter starts off disrespectfully. An assumption is made that all female circumcision is the same (and as disfiguring and disabling as the worst described examples). It is also assumed that cultural practices are fixed and unchanging, and that, because disfiguring and disabling procedures were performed in the past, people from that culture will wish to continue with the same practice forever. Essen [44] provides a comprehensive discussion of this topic and particularly highlights the strange disparity where genitoplasty for body dysmorphic disorder is acceptable in western societies even though it directly transgresses legislation passed on female genital mutilation.

This example illustrates well the benefits of understanding the cultural background in the presence of disagreement. I doubt very much that the surgeons expected at the outset that such a minor procedure would be satisfactory to the family. The benefit of considering spiral dynamics is also well illustrated. Had this family been in Somalia, their very survival would probably depend on remaining a part of the family. Had this mother and daughter refused to comply with cultural norms, they may well have been subject to violent coercion or, at worst, excommunication. Given that there is no provision for welfare payments, free health care, or social housing in Somalia and that the only security comes from membership of the community, this would be a dire prospect. In the USA, this is different. The option of leaving the Somali community is not as dire. Therefore, considering transgressing cultural norms is a possibility. The safety net of the USA means that considerations other than just survival can be considered.

Had the surgeons in Seattle been allowed to proceed, you could imagine a number of outcomes. The Somali Community may have had increased trust in the institution (compared to if there had been a blunt refusal), which would make them more likely to seek care. The option of having a "ritual nick" would have been very welcome to those Somalis who had begun to be Americanised, giving them the option of remaining within the Somali community by conforming to those norms without causing harm to their daughters. Refusing to perform a procedure does not mean it will not happen, it just means that if it happens, it is likely to be more harmful to the girl. There are parallels here with abortion. Making abortion illegal does not stop it from happening. However, it means that those women having an abortion are more likely to die. So how would I respond if a Somali woman asked

me (as a GP in New Zealand) to circumcise her daughter? First of all, I would thank her for trusting me sufficiently to even discuss the issue as I know that many would not. I would ask her to describe in detail exactly what she wanted and the reasons why. I would explain to her that the law in New Zealand prohibits me from performing such a procedure and so, irrespective of what I thought about it, I would not be willing to perform it for fear of jeopardising my career. I would discuss with her what options she would consider (would she be planning to go back to Kenya for the procedure, for example) and I might describe my fears of the consequences of that course of action in much the same way as I discuss my fears regarding patients who go to Thailand for cosmetic procedures who may not understand the risks of exposure to multi-resistant bacteria, their lack of any recourse if things go wrong, and their inability to judge the quality of the clinic they might attend. I would reassure the mother that, no matter what she decided, I would continue to be their GP and, in particular, if she did go overseas to have the procedure, I would be prepared to provide care on their return. I could choose, as some US Paediatricians do in relation to vaccination [45], to refuse to care for the girl if her mother does not agree with my advice. However, my view is that building trust and maintaining a relationship is the best way to provide the best outcome for the daughter. If I refuse to care for her after she has had the procedure, I have no ability to affect the outcome.

As a politically active doctor, I would continue to engage in debates surrounding the development of New Zealand law to try to allow it to reflect our increasingly diverse population instead of just reflecting the dominant majority. Current New Zealand law and practice allows "genital mutilation" of boys and intersex children for cultural or cosmetic reasons despite banning any genital surgery on girls. A significant risk is that the majority culture will criticise the practices of incoming minorities without applying the same principles to their own practices.

## 4. Conclusion

A (bio)ethics based solely on an analysis to determine the right course of action predicated on the idea that there is such a thing as the "best" approach has significant limitations in addressing problems at the individual clinical level, the national level, and the international level. An ethics process that pays more attention to the methods used to decide how to move forward in the presence of a difference of opinion is needed. While debating what a normative ethics ought to be has value, at a practical level, descriptive ethics may be more useful, so that the parties in the midst of a disagreement can have a better understanding of why they disagree. Attention to both a cultural background and their level of psychological development can help with this understanding. At all these levels of society, a clear understanding of how to achieve workable compromises would seem to be the least of what is needed but more work is required.

At a clinical level, the process of negotiating an agreed management plan is time and resource intensive and there will be many occasions when a good enough outcome will be reached by a more paternalistic approach. In what circumstances is this acceptable? There is such a thing as a bad compromise. The obvious international example being Neville Chamberlain's announcement of "Peace for our Time" on 30 September, 1938, which was a signed agreement with Adolf Hitler that turned out to be worse than worthless. How do we know the difference between a good and bad compromise? It is equally true that "sticking to your principles" can also lead to bad outcomes, as demonstrated by the children who received no care at all as they died, because the doctors caring for them insisted on treatment that the parents did not want [22]. There will always be value in discussing and debating what the right thing to do is. The ongoing debate around human rights that began after World War 2 has undoubtedly contributed considerably to the improvement in human rights, even if countries continue to disagree (particularly by their behaviour if not their rhetoric) with the detail. Working to live together in a pluralistic society is not easy. It is hard to respect and not to judge the person and focus instead on the behaviour. It is uncomfortable living with people you disagree with. Do I really want to live in a society that legalises euthanasia, or accepts large levels of income disparity, or values people who do not believe in climate change? For the large international issues, there is no other

alternative but to work with people that we do not like or do not agree with if we want to address problems of pandemics, climate change, failed states, refugee movement, and famines. For many of us at a national level, we no longer have the choice of not living in a pluralist society. At a community level, it is possible to limit the diversity of the people that we live among, but it may be better to focus on and highlight the benefits of pluralism/diversity and challenge those who suggest that living in such isolation is desirable. The challenges we are facing are unlikely to be managed by maintaining the status quo.

**Funding:** This research received no external funding.

**Conflicts of Interest:** The author declares no conflict of interest.

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
