# Peer review of "(Bio)Ethics in a Pluralistic Society"

_challenges, doi:10.3390/challe10010012_

Round 1

Reviewer 1 Report

This is clearly written. I would like the author to have spent time explaining how people who hold different beliefs, informed by different values, can reach compromise or consensus. Understanding the premises for the conclusions is important, but insufficient. The author may want to consider the language of cultural humility rather than competence. 

Author Response

I have included clarification of the relatioinship between the concepts of cultural competence and cultural humility, and yes the language of cultural humility is useful.

I have included two specific examples on how to reach compromise to illustrate these concepts.

Reviewer 2 Report

The proposed approach, which is based on the so called ethics of compromise, is sound and well analysed. Ethics of compromise is certainly fruitful, when different values and personal choices  (in cliinical context, between doctor and patient) are at stake.

Two main questions should be addressed.

First, it is not clearly discussed the interplay between the ethics of compromise and the principle of informed consent: how should the compromise be reached if the patient withdraw or withhold the consent?

Second, the main challenges in pluralistic bioethics arise from the conflict between values and traditions of minority groups (recently settled groups, migrants, religious or ethno-cultural minorities), and values and traditions embedded in law and in mainstream practices. This question seems to be completely unaddressed.

Professionals codes of conduct, laws, national and international legal instruments aren't (as the author rightly recognises) ethically neutral: they express values and principles, which are not only culturally specific, but are also binding. To what extent a compromise is possible, in these cases?

Author Response

Thankyou for the helpful review I have expanded the section on informed consent and added a section on value conflict with minority populations

Round 2

Reviewer 2 Report

The article is now ready for publication, the Author has adequately answered to the remarks and the comments.